# Comparison of the Vertical Force Distribution in the Paws of Dogs with Coxarthrosis and Sound Dogs Walking over a Pressure Plate

**DOI:** 10.3390/ani10060986

**Published:** 2020-06-05

**Authors:** Jane P. L. Moreira, Alexander Tichy, Barbara Bockstahler

**Affiliations:** 1Department of Companion Animals and Horses, Small Animal Surgery, Section of Physical Therapy, University of Veterinary Medicine Vienna, 1210 Vienna, Austria; Barbara.Bockstahler@vetmeduni.ac.at; 2Bioinformatics and Biostatistics platform, University of Veterinary Medicine Vienna, 1210 Vienna, Austria; Alexander.Tichy@vetmeduni.ac.at

**Keywords:** coxarthrosis, dogs, gait analysis, paw, pressure plate, vertical force distribution

## Abstract

**Simple Summary:**

The study of biomechanics for dogs with coxarthrosis is an important tool for diagnosis and treatment evaluation. Seeking a better view of the load distribution during the gait in dogs with coxarthrosis, we used a pressure plate to measure the vertical forces in the paws. The results suggested that walking dogs with coxarthrosis redistributed the load mainly to the caudal quadrants of the paws of the unaffected limbs. The performed methodology is another new possibility for the evaluation and clarification of biomechanical events in the course of coxarthrosis.

**Abstract:**

In the present study, we used a pressure plate to investigate the ground reaction forces of limbs and the vertical force distribution (VFD) within the paws of dogs with coxarthrosis. We included 23 sound dogs (G_Sou_) and 23 dogs with hip osteoarthrosis (G_Cox_). The dogs walked over a pressure plate and the peak vertical force (PFz), vertical impulse (IFz) as the percentage of the total force, and time of occurrence of PFz as a percent of the stance phase duration (TPFz%) were evaluated, as well for the entire limb as in the paws (where the paws were divided into four quadrants). The G_Cox_ presented a lower PFz% in the lame hind limb than in others and transferred the weight to the caudal quadrants of the front limbs. IFz% was lower in the lame limb and was counterbalanced through higher loading of the caudal quadrants in all unaffected limbs. TPFz% was reached later in the lame limb than in the contralateral limb and the G_Sou_, specifically in the caudomedial quadrant. In conclusion, we found complex compensatory effects of lameness in the hind limb, and this methodology was useful to define the VFD within the paws of dogs.

## 1. Introduction

The most common cause of chronic pain in dogs is osteoarthrosis [1]. Thus, when a coxarthrosis occurs, the development of diagnostic studies and treatment alternatives is important to minimize the consequences and to reduce the costs of drugs and therapies. Joint diseases in the hind limbs of dogs are most responsible for musculoskeletal alterations, and are consequently related to biomechanical adjustments in the gait and redistribution of loading on the limbs [2]. In the clinical routine, the subjective assessment of the hip is based on image exams and orthopedic examinations [3]. Kinetic and kinematic studies, which help to understand the pattern of gait and the vertical force distribution (VFD) between the limbs. The study of kinetics and kinematics in animals has been developing and improving in recent decades. In kinetics, several methodologies are used to demonstrate and validate the distribution of ground reaction forces (GRF) in healthy dogs and dogs with orthopaedic conditions [2,4,5,6,7,8,9,10,11,12,13]. The most common techniques for gait analysis are force plates (FP) and pressure plates (PP) [9,14]. While the use of FP is recognized as the gold standard [8,12,13,15,16,17,18,19], the use of PP has become increasingly evident in recent years [9,12,13,14,18,19,20,21,22,23,24], to determine and compare the pressure distribution between limbs [18,19,25,26], as well as the analysis of the VFD on the paws and pads of the sound [12,20,21,22] and diseased animals [13,21].

The literature mostly shows that the vertical GRF are significantly lower in affected hind limbs [1,4,11,27,28,29,30,31], and biomechanical adaptations and gait pattern alterations are also expected in the limbs that are not affected to compensate and redistribute the load of body weight [2]. Budsberg et al. [1] proposed that the redistribution happened in trotting dogs with unilateral chronic coxarthrosis. Kennedy et al. [3] suggested that the VFD occurred predominantly by transferring force between sides of trotting pairs instead of to front limbs. Katic et al. [32] compared the conventional GRF analysis with Fourier analysis. Those authors found that a unilateral degenerative joint disease of the hip also influenced the GRF of the front limbs in dogs walking on a treadmill equipped with four FP. Unlike studies on the GRF of the limb as a whole, there is still no information on the distribution of pressure or the distribution of force in the pads in dogs with coxarthrosis.

In the present study, we used a PP to clarify the GRF of limbs and the VFD within the paws of dogs with coxarthrosis. We expected that animals with coxarthrosis would show decreased vertical forces in the cranial/caudal and lateral/medial regions of the affected paws compared to sound animals and that the distribution of force between the paws would change, comparing the non-affected limbs of diseased dogs with same paws of sound dogs.

## 2. Materials and Methods

The data used in this retrospective study were obtained from the database of the University of Veterinary Medicine Vienna. The measurements were collected between the years 2013–2018 and were discussed and approved by the institutional ethics and animal welfare committee of this university under the Good Scientific Practice guidelines and national legislation (Approval No. ETK—01/03/2017, ETK—05/09/2016, ETK—09/12/2015, and ETK—04/05/97/2013). The owners of the animals signed the written acceptance for study inclusion.

### 2.1. Dogs and Inclusion Criteria

To perform this study, 44 dogs were divided into two groups. The group sound (G_Sou_) was formed by 23 healthy dogs (six Golden Retrievers, five mixed breeds, three Labradors, two Border Collies, two Wolfdogs, one Groenendael, one Rottweiler, one German Shepherd, one Dachshund, and one German Shorthaired Pointer). In the G_Sou_ there were six females, nine males, four spayed females, and three neutered males, with a mean age of 4.3 ± 2.3 years and a mean body mass (BM) of 25.4 ± 9.1 kg. To determine their good health status, the dogs underwent physical, neurological, and orthopedic examinations.

The inclusion criteria in G_Sou_ included no signs of lameness, pain, musculoskeletal, or neurological abnormalities. Sound dogs were carefully chosen from an available databank and were selected to match, as closely as possible, to either breed or phenotypic characteristics of dogs with coxarthrosis. 

The group coxarthrosis (G_Cox_) was formed by 23 dogs (seven mixed breeds, three Golden Retrievers, three Labradors, three German Shepherds, two Rottweilers, one Appenzeller Sennenhund, one Tibetan Terrier, one Samoyed, one Border Collie, and one Vizsla). In the G_Cox_, there were one female, six males, eight spayed females, and seven neutered males, with a mean age of 8.5 ± 3.3 years, with a mean BM of 27.5 ± 11.6 kg, with coxarthrosis in at least one hip articulation. The inclusion criteria in this group were the existence of clinical signs and compatible images (X-rays and/or computed tomography) with coxarthrosis at least in one hip articulation, a clinical unilateral hind limbs (HL) lameness, and the absence of pain and/or orthopedic alterations in the front limbs (FL). 

To substantiate the lameness that was already clinically determined, a symmetry index (SI%) was also used as an inclusion criteria. To be included in this study, sound dogs should present an SI% for peak in vertical force (PFz) and vertical impulse (IFz) lower than 3% in front and hind limbs [33]. In G_Cox_, dogs should present an SI% for PFz and IFz for the HL higher than 4%. 

### 2.2. Measurement Procedures

For measurements, the dogs walked over a calibrated Zebris FDM Type 2 pressure plate (Zebris Medical GmbH, Allgäu, Germany), in size 203.2 × 54.2 cm, with a sampling rate of 100 Hz. The plate was installed in a 7 m runway, covered with a 2 mm thick rubber mat. The data were stored with WinFDM software (v 1.2.2; Zebris Medical GmbH) and processed by the software Pressure Analyzer 1.3.0.2 (Michael Schwanda, Königstetten, Austria). The dogs were guided by their instructed owners, at a comfortable velocity (G_Sou_ = 1.14 ± 0.16 m/s; G_Cox_ = 1.06 ± 0.18 m/s) and acceleration (G_Sou_ = 0.012 ± 0.06 m/s2; G_Cox_ = 0.012 ± 0.05 m/s2). At least five valid trials were collected for each dog. All trials were recorded by a Panasonic NV-MX500 camera, to be able to match the footprints tot the correct extremity.

To calculate the SI%, the following formula, modified from Budsberg et al. [33], was used
(1)$$SIXFz=abs\left(\frac{XFzLF-XFzRF}{XFzLF+XFzRF}\right)\times 100$$
where *XFz* = value of PFz% or IFz%, *abs* = absolute, *LF* = left front limb, and *RF* = right front limb. The same formula was used to calculate the SI (%PFz or %IFz) for the HL. The result is given in percent. 

### 2.3. Evaluated Parameters in the Entire Limb

The evaluated GRF parameters for the entire limb were PFz in Newtons and IFz in Newtons/second, both normalized as percentual of total force and abbreviated as PFz% and IFz%, respectively, stance phase duration in seconds (SPD (s)) and time of occurrence of peak in vertical force as a percent of the stance phase duration (TPFz%). To compare limbs, the following nomenclature was used in G_Sou_: right hind limb (RH), right front limb (RF), left hind limb (LH), and left front limb (LF). In G_Cox_, according to the clinically demonstrated lameness, one of the hind limbs showed lower values for PFz and IFz; this limb was subsequently referred as the lame limb (L). To enable statistical analysis, this limb was then always compared with the right hind limb of sound dogs. The right front limb was seen as ipsilateral (IPSI), the left hind limb as contralateral (CONT), and the left front limb as diagonal (DIA). The abbreviations are summarized in Figure 1.

### 2.4. Evaluated Parameters in the Paws

To delimit the quadrants, the software calculated the midpoint of the maximum length in the cranial/caudal and medial/lateral directions of each paw print, obtaining equal quadrants anatomically denominated craniomedial (CraMe), craniolateral (CraLa), caudomedial (CauMe), and caudolateral (CauLa). The abbreviations are summarized in Figure 2. Parameters under investigation were the same as for the entire limb, except SPD.

To analyze the VFD in the quadrants, the PFz% and IFz% values were normalized to the total force and presented as % (i.e., PFz% and IFz%), in which the sum of PFz% and IFz% of the 16 quadrants was equal to 100%. We used formula [12], presented below
(2)$$TF_{nk}\left(\%\right)=\frac{100\times X_{nk}}{\sum_{k=1}^{4}\sum_{n=1}^{4}X_{nk}}$$
where *X* represents PFz% or IFz%, *n* a limb (LF/DIA, RF/IPSI, LH/CONT, and RH/L) and *k* for one quadrant (CraLa, CraMe, CauLa, and CauMe).

### 2.5. Data Analysis

The data were processed by SPSS software, version 24. A normal distribution was confirmed by the Kolmogorov–Smirnov test. A mixed model was used to analyze the limbs (LH/CONT versus RH/L, LF/DIA versus RF/IPSI), direction (front versus hind), and quadrants (cranial versus caudal, lateral versus medial). A general linear model (ANOVA) was used to calculate the differences between groups. In each paw, the cranial-lateral or medial-quadrants were compared with the caudal-lateral or medial-, respectively, and lateral-cranial or caudal-quadrants with the medial-cranial or caudal-, respectively. Each quadrant was compared with its correspondent in the contralateral limb (front or hind limb) within the group. Between groups, each quadrant was compared with its corresponding paw from the other group [13]. Multiple comparisons were performed by applying Sidak’s alpha correction procedure. The data are presented as mean ± standard deviation (SD) and the level of significance was set at *p* < 0.05.

## 3. Results

### 3.1. Symmetry Index

The values of the symmetry index (%), as well as the p-values, are presented in Table 1. The SI (% PFz) and SI (% IFz) were significantly higher in G_Cox_ than in G_Sou_ for both the front and hind limbs.

### 3.2. GRF—Limbs

In this section, the results from the entire limbs within and between groups will be described.

#### 3.2.1. Comparison Within Groups

G_Sou_: For all analyzed variables, in the G_Sou_, neither differences between the contralateral FL or the contralateral HL were observed. When comparing the right and left front limbs with the respective hind limbs, significant differences were observed for all parameters. In the FL, the PFz% and IFz% values were higher, the SPD (s) was longer, and the TPFz% was achieved later than in the HL.

G_Cox_: In G_Cox_ no significant difference was observed between the DIA and IPSI limbs (i.e., both FL) in all parameters under investigation. The PFz% and IFz% values were significantly higher in CONT than in the L limb. The SPD was significantly longer and the TPFz% was achieved significantly earlier in CONT than in the lame limb. Significantly higher values were observed in both FL than in both HL for PFz%, IFz% and for TPFz%. For the SPD this was also true, except when comparing the CONT and IPSI limbs, where the values were not statistically different. The values of PFz%, IFz%, SPD (s), and TPFz%, as well as the *p* values from both groups, are presented in Table 2 and Table 3, respectively, and summarized in Figure 3 and Figure 4.

#### 3.2.2. Comparison Between Groups

The values of PFz%, IFz%, SPD (s) and TPFz%, as well as the *p* values comparing groups, are presented numerically in Table 2 and Table 4, respectively. In the right hind limbs of sound dogs (RH, G_Sou_), higher significant values for PFz%, IFz%, and an earlier occurrence of PFz% were observed when compared to G_Cox_. In contralateral limbs, G_Cox_ showed a higher IFz% than G_Sou_. Comparing the ipsilateral limb between groups, in G_Cox_, PFz% was significantly higher than in G_Sou_. No significant differences were found between the diagonal limbs. 

### 3.3. GRF—Quadrants

Comparing the results for PFz% and IFz% of the individual quadrants within the paws, the results are as follows:

In G_Sou_, both PFz% and IFz% in the cranial quadrants were significantly higher than in the caudal quadrants in all limbs. In the G_Cox_, this was also true for IFz%. However, for PFz% values, it was also true for the hind limbs and the CraMe versus the CauMe quadrants in the front limbs. In this group, the lateral quadrants of the FL were equally loaded. In the case of the diagonal paw, this was due to a significant reduction in the PFz% from the CraLa quadrant, and in the ipsilateral paw, due to a significant increase in the CauLa quadrant, both differences when compared to the healthy group.

If we compare the lateral quadrants with the respective medial quadrants, in both groups, for PFz% the CraLa quadrants did not differ in principle from the CraMe results, except in the diagonal extremity (i.e., LF) of healthy dogs. In contrast, the IFz% in G_Sou_ showed ever higher values of the CraLa than in the CraMe quadrants, except for the CraLa versus CraMe of the RH.

In all four paws of dogs in G_Cox_, the CraLa and CraMe IFz% values did not differ. In the FL, this resulted from a significant reduction in the IFz% in the CraLa quadrants when compared to the respective ones in the G_Sou_. In the area of the lame paw, there was a significant reduction in the IFz% in both cranial quadrants compared to G_Sou_. It is noticeable that in G_Cox_, the PFz% and IFz% of the CauMe quadrant of the lame limb were significantly lower than in the corresponding quadrant of the contralateral limb. The IFz% of the CauLa quadrants also increased significantly in the contralateral limb when compared to the corresponding quadrant of the lame limb. In both groups, the CauLa quadrants of the FL had a significantly higher PFz% and IFz% than the CauMe quadrants. In the HL, the animals of both groups also showed significantly higher values for IFz% in both CauLa quadrants than in CauMe. In PFz%, this only applied to the right hind extremity (i.e., L in G_Cox_) in G_Sou_. In G_Cox_, there was no difference between the quadrants. These results are shown in Table 5 and Table 6.

Comparing groups, in the L from G_Cox_, values of IFz% in both cranial quadrants were lower than in the corresponding limb in G_Sou_. At the same time, there was an increase in IFz% in the caudal quadrants of the contralateral limb. In the lame limb, the PFz% value of the CraLa quadrant was significantly lower in the G_Cox_ than in G_Sou_. In both FL, the IFz% of the caudal quadrants was significantly higher in G_Cox_ than in G_Sou_. Higher PFz% was also observed in G_Cox_ than in G_Sou_, but only in the CauMe quadrants of diagonal limb and CauLa of the ipsilateral limb. 

As well in the FL, significant differences were observed in the IFz% values in the CraLa quadrants, but with higher values in G_Sou_ than in G_Cox_. For PFz%, a significantly higher value was observed in the CraLa quadrant of the diagonal limb, and also in G_Sou_. The TPFz% was also measured and was reached significantly earlier in the caudal than in the cranial quadrants in all limbs from both groups. In the FL, the TPFz% was reached significantly earlier in the CraLa than in the CraMe quadrants, also in both groups. When comparing groups, the TPFz% was reached significantly later in the CauMe quadrant in the lame limb from G_Cox_ than in the respective quadrant in G_Sou_. These results are summarized in Figure 5, Figure 6 and Figure 7, and the *p* values are listed in Table 7.

## 4. Discussion

Our hypotheses that animals with coxarthrosis would present changes in the VFD between limbs and quadrants when compared to healthy animals were confirmed. There was a redistribution of forces, not only between limbs but also between quadrants, when compared to the healthy group.

In the G_Sou_, higher values of PFz% and IFz% in the front compared to the hind limbs were expected and consistent with the available literature [2,7,9,10,12,15,32,34,35,36,37]. The SI (%Pfz and %IFz) for this group also corresponded to those demonstrated by other studies [1,2,17,35,37], confirming the lameness-free status. The SPD (s) in FL was significantly longer and, the TPFz% was reached later in FL than in LH, agreeing with the results found by Schwarz et al. [12].

Comparing the lame and contralateral limbs from G_Cox_, the significantly lower PFz% and IFz% values in the lame limb were expected, as demonstrated in studies that used force [2,3,32,38,39] and pressure plates [11,26]. A significant increase in the PFz% value for the ipsilateral limb in G_Cox_ was also observed when compared to the group of healthy animals, but no significant difference in the IFz% values. For the IFz, the results agree with another study [3], where no redistribution of IFz% to the FL in dogs with coxarthrosis was observed. PFz% is a variable with good sensitivity and specificity in the evaluation of GRF in PP [9,40], indicating that there was also a weight redistribution to the ipsilateral thoracic limb, as also shown by Katic et al. [32] via Fourier analysis, and by Fischer et al. [2], although the last one was carried out on animals with induced hind lameness.

Despite all the similarities and disagreements observed between the results of this study and the available literature, the differences in the methodologies (FP, integrated FP on a treadmill, or PP), affected joints (tarsal, knee, and hip), and type of lameness (acquired or induced) interfere in the comparison of results and their discussion. In the present study, we used a PP, and the majority of the published studies to date used one or more FP. The use of a PP to determine GRF in dogs is considered valid [9,24,40] and also has some advantages over single FP, such as the record of multiple, successive, simultaneous, and collateral foot contact during walking, demanding fewer trials with high reproducibility [9,14,40,41,42]. We used the same methodology described in other works [12,13,22], in which the paws were divided into quadrants. This allowed us to evaluate the VFD in the paws of animals of different sizes with lower plate resolution. Further studies using plates with a higher resolution are recommended.

The higher load in the cranial than in the caudal quadrants in the paws of sound animals is following previous studies that measured either the pressure [43] or the GRF in each paw pad separately [20,21], as well as with studies that used the same methodology as us, dividing the paws into quadrants [12,13]. In agreement with the results published by Braun et al. [13] in sound dogs, the PFz% and IFz% values of the CauLa quadrants were significantly higher than in the CauMe quadrants in all limbs, and the IFz% values of the cranial quadrants of the front limbs were also higher than the caudal quadrants in the same limbs in the G_Sou_. From this, it is possible to conclude that, in healthy animals, the GRF tended to be higher in the cranial and lateral quadrants of all limbs. In animals from G_Cox_, the difference in PFz% between the lateral quadrants in both front limbs disappeared due to a significant decrease in the CraLa quadrant from limb DIA and an increase in the quadrant CauLa in the limb IPSI. As the IFz% also presented differences in both the caudal and CraLa quadrants from both FL, we can deduce that there was a transference of weight to the FL, especially to caudal quadrants. Using different methodologies, Souza et al. [44] found similar results in Pitbulls with cranial cruciate ligament rupture, where the dogs compensated the loading on the metacarpal and metatarsal pads from the not affected limbs as could be compared with the caudal quadrants. 

In a study with dogs with elbow osteoarthritis [13], the same pattern was demonstrated, where the GRF were generally higher in the caudal quadrants of unaffected limbs. In the present study, there was a clear redistribution of the load within the paw to the unaffected limb (CONT), both when comparing groups and comparing quadrants within G_Cox_ itself. Thus, we can deduce that, when compared to healthy animals, dogs with coxarthrosis tended to shift the weight mainly to the caudal quadrants of the unaffected limbs and to reduce the load on the cranial quadrants of the limb with coxarthrosis (L).

The results obtained for TPFz% agreed with the previous studies [12,13] regarding the relationship between the front and hind limbs and the cranio/caudal quadrants and according to Braun et al. [13], regarding the lateral quadrants of the front limbs. The only difference found between groups for this variable allowed us to infer, along with the other results, that the CauMe quadrant of the L limb was the one that undergoes the greatest changes in the VFD of dogs with coxarthrosis. The morphology, manner of moving, body size, shape, and weight directly affected the values of GRF [7,15,36,45,46,47]. The sample groups of this study were, however, heterogeneous, which could affect the analysis and interpretation of results. To reduce the impact of this heterogeneity, we chose sound dogs that could be approximately morphologically similar to dogs with orthopaedic conditions and normalized the absolute values to a percent of the total force. This was, of course, only possible to a certain extent. This can be seen in the high SD of the body mass in both groups. This was a limitation of the study which must be taken into account when interpreting the results. To confirm our results, further studies should be conducted with dogs of the same breed.

The distribution of forces within paws in the group of sound animals was compatible with the results of Schwarz et al. [12] and this allowed us to have more confidence in the interpretation of the obtained data.

It has already been shown that radiographic signs in dogs with orthopedic alterations in the hip joint do not match with the subjective evaluation of pain performed by veterinarians and owners [48]. In dogs with gonarthrosis, radiographic signs and GRF obtained from an FP were compared and no correlation was observed between images and the severity of lameness and limb function [49]. On the other hand, using PP, it was demonstrated that the higher the hip dysplasia degree was in German shepherd dogs, the lower the PFz was for the affected limb [11]. 

In the present study, we observed that the GRF within the paws showed accurate details regarding the load distribution between and within the paws. Thus, we can suggest that the use of the performed methodology favors the study of orthopedic alterations in dogs as it allows the detailing of VFD within the paw with good sensitivity. Further studies incorporating the X-ray results should also be carried out and, consequently, the treatment and revaluation of these cases would become easier and more objective, benefiting both the animals and their owners and even veterinarians. Further, the data used here were from animals with coxarthrosis due to different causes that were not specified. More specific and delimited studies with different causes of coxarthrosis should be performed.

## 5. Conclusions

This work provided an overview of the VFD within the paws of dogs with coxarthrosis. Dogs with coxarthrosis presented changes in the VFD between limbs and quadrants when compared to sound animals, tending to transfer the load to the contralateral hind limb and the ipsilateral front limb. Within the paws, the dogs with coxarthrosis tended to transfer the load mainly to the caudal quadrants of the unaffected limbs and to reduce the load on the cranial quadrants of the affected limb. The performed methodology is another new possibility for the evaluation and clarification of biomechanical events in the course of coxarthrosis.

## Figures and Tables

**Figure 1 animals-10-00986-f001:**
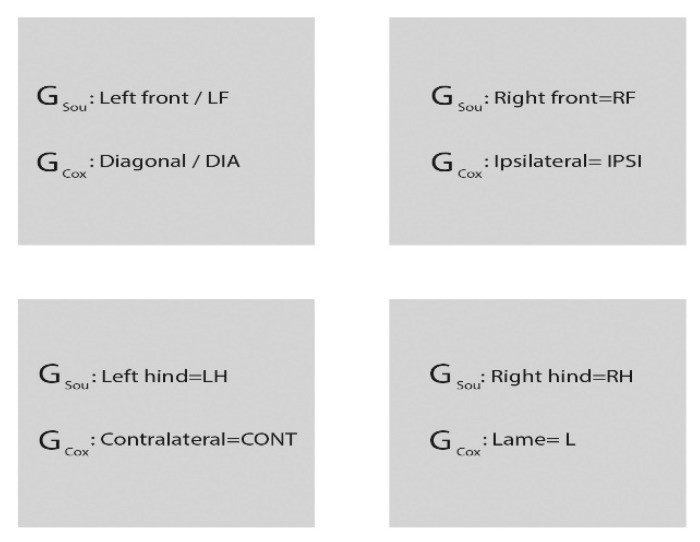
The G_Sou_ and G_Cox_ dog limbs nomenclature scheme. G_Sou_ (group sound); G_Cox_ (group coxarthrosis); LF (left front limb); RF (right front limb); LH (left hind limb); RH (right hind limb); DIA (diagonal limb); IPSI (ipsilateral limb); CONT (contralateral limb); L (lame limb).

**Figure 2 animals-10-00986-f002:**
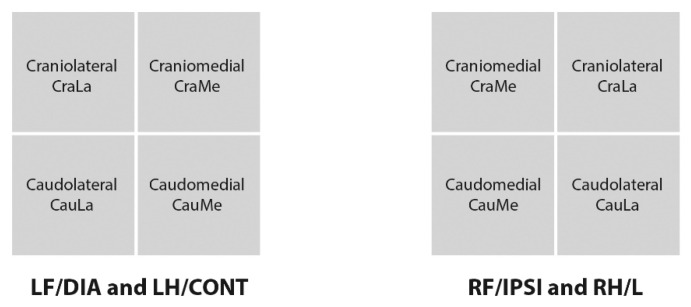
Quadrants nomenclature scheme. LF (left front limb); RF (right front limb); LH (left hind limb); RH (right hind limb); DIA (diagonal limb); IPSI (ipsilateral limb); CONT (contralateral limb); L (lame limb).

**Figure 3 animals-10-00986-f003:**
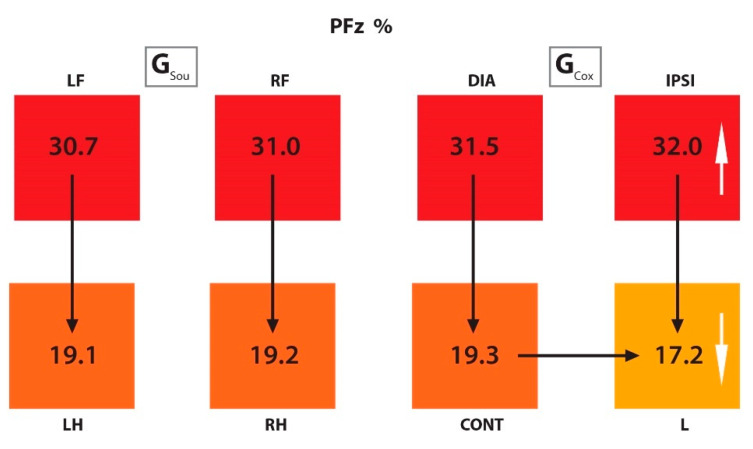
Scheme representing the PFz% of the limbs from G_Sou_ and G_Cox_. The black arrows represent a significant difference between the limbs within groups (for example, in G_Cox_, there was a difference between CONT and L limbs), and the white arrows, between the groups (for example, there was a difference between G_Sou_ and G_Cox_ in RF/IPSI and RH/L limbs). The color scale within each group: higher values are represented in darker colors and lower values in lighter colors. PFz% (peak of vertical force normalized as percentual of total force); G_Sou_ (group sound); G_Cox_ (group coxarthrosis); LF (left front limb); RF (right front limb); LH (left hind limb); RH (right hind limb); DIA (diagonal limb); IPSI (ipsilateral limb); CONT (contralateral limb); L (lame limb).

**Figure 4 animals-10-00986-f004:**
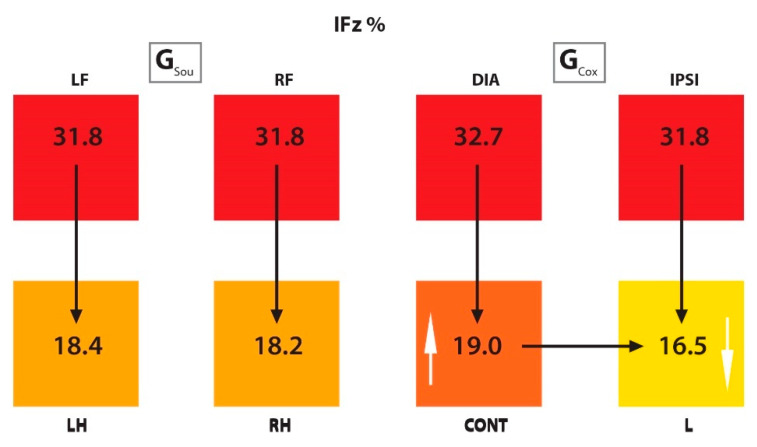
Scheme representing the IFz% of the limbs from G_Sou_ and G_Cox_. The black arrows represent a significant difference between limbs within groups (for example, in G_Cox_, there was a difference between CONT and L limbs), and the white arrows, between groups (for example, there was a difference between G_Sou_ and G_Cox_ in LH/CONT and RH/L limbs). The color scale within each group: higher values are represented in darker colors and lower values in lighter colors. IFz% (vertical impulse normalized as percentual of total force); G_Sou_ (group sound); G_Cox_ (group coxarthrosis); LF (left front limb); RF (right front limb); LH (left hind limb); RH (right hind limb); DIA (diagonal limb); IPSI (ipsilateral limb); CONT (contralateral limb); L (lame limb).

**Figure 5 animals-10-00986-f005:**
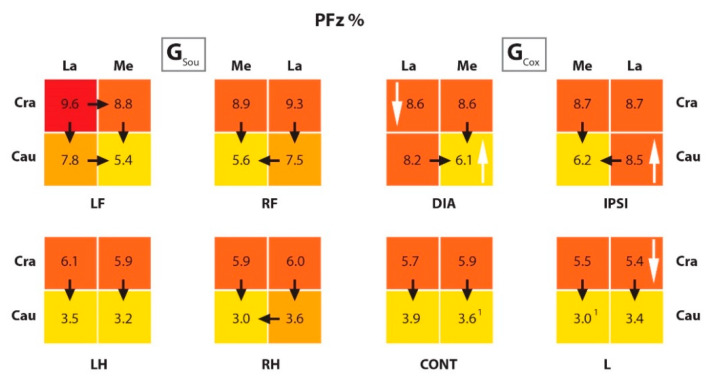
The scheme representing the PFz% in quadrants of the paws from G_Sou_ and G_Cox_. The black arrows show a significant difference between the quadrants within paws (for example, in the LF limb from G_Sou_ there was a difference between the CraLa and CraMe quadrants) and the white arrows, between groups (for example, there was a difference between the CraLa quadrants from the LF limb in G_Sou_ and DIA limb in G_Cox_); and the numbers (^1^), between equivalent quadrants in different limbs within a group (for example, in the G_Cox_, there was a difference between the CauMe quadrants from the CONT and L limbs). The color scale within paws: higher values in darker colors and lower values in lighter colors. PFz% (peak of vertical force normalized as percentual of total force); G_Sou_ (group sound); G_Cox_ (group coxarthrosis); LF (left front limb); RF (right front limb); LH (left hind limb); RH (right hind limb); DIA (diagonal limb); IPSI (ipsilateral limb); CONT (contralateral limb); L (lame limb); Cra (cranio); La (lateral); Cau (caudo); Me (medial).

**Figure 6 animals-10-00986-f006:**
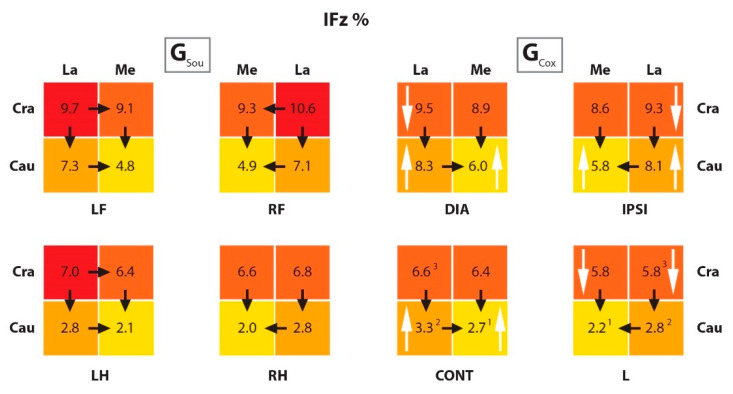
The scheme representing the IFz% in quadrants of the paws from G_Sou_ and G_Cox_. The black arrows show a significant difference between the quadrants within paws (for example, in the LF limb from G_Sou_ there was a difference between the CraLa and CraMe quadrants); the white arrows, between groups (for example, there was a difference between the CraLa quadrants from the LF limb in G_Sou_ and DIA limb in G_Cox_); and the numbers (^1 2 3^), between equivalent quadrants in different limbs within a group (for example, in the G_Cox_, there was a difference between the CauMe quadrants from the CONT and L limbs). The color scale within paws: higher values in darker colors and lower values in lighter colors. IFz% (vertical impulse normalized as percentual to the total force); G_Sou_ (group sound); G_Cox_ (group coxarthrosis); LF (left front limb); RF (right front limb); LH (left hind limb); RH (right hind limb); DIA (diagonal limb); IPSI (ipsilateral limb); CONT (contralateral limb); L (lame limb); Cra (cranio); La (lateral); Cau (caudo); Me (medial).

**Figure 7 animals-10-00986-f007:**
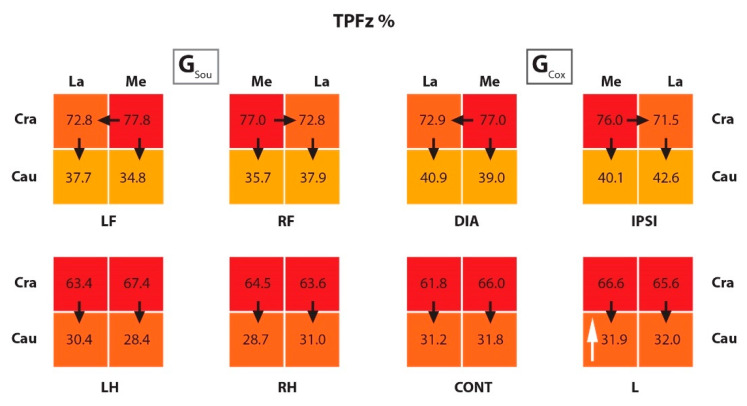
The scheme representing the TPFz% in the quadrants of the paws from G_Sou_ and G_Cox_. The black arrows show a significant difference between quadrants within paws (for example, in the LF limb from G_Sou_ there was a difference between the CraLa and CraMe quadrants); and the white arrow, between groups (for example, there was a difference between the CauMe quadrants from the RH limb in G_Sou_ and L limb in G_Cox_);. The color scale within paws: higher values in darker colors and lower values in lighter colors. TPFz% (time of occurrence of peak of vertical force as a percent of the stance phase duration); G_Sou_ (group sound); G_Cox_ (group coxarthrosis); LF (left front limb); RF (right front limb); LH (left hind limb); RH (right hind limb); DIA (diagonal limb); IPSI (ipsilateral limb); CONT (contralateral limb); L (lame limb); Cra (cranio); La (lateral); Cau (caudo); Me (medial).

**Table 1 animals-10-00986-t001:** SI (%PFz) and SI (%IFz) on the front limbs and hind limbs ± standard deviation.

SI%	G_Sou_	G_Cox_
SI (%PFz)_FL	0.97 ± 0.86 ^a*^	2.09 ± 2.15
SI (%PFz)_HL	1.36 ± 1.09 ^a^	5.83 ± 3.32
SI (%IFz)_FL	1.43 ± 1.13 ^a^	3.37 ± 3.21
SI (%IFz)_HL	1.62 ± 1.22 ^a^	7.45 ± 3.16

a = significant difference between G_Sou_ and G_Cox_ at a significance level at *p* < 0.001. a* = significant difference between G_Sou_ and G_Cox_ at a significance level at *p* < 0.05. G_Sou_ (group sound); G_Cox_ (group coxarthrosis).

**Table 2 animals-10-00986-t002:** Mean values ± standard deviation of the PFz%, IFz%, SPD (s), and TPFz% in the IPSI/RF, DIA/LF, CONT/LH, and L/RH limbs from dogs of G_Sou_ and G_Cox_.

**Groups**	**PFz%**			
	DIA (LF)	IPSI (RF)	CONT (LH)	L (RH)
G_Sou_	30.73 ± 0.3	30.99 ± 0.6	19.07 ± 0.31	19.2 ± 0.31
G_Cox_	31.45 ± 0.38	32.02 ± 0.38	19.32 ± 0.33	17.21 ± 0.36
	**IFz%**			
	DIA (LF)	IPSI (RF)	CONT (LH)	L (RH)
G_Sou_	31.8 ± 0.26	31.78 ± 0.3	18.24 ± 0.25	18.17 ± 0.26
G_Cox_	32.74 ± 0.45	31.77 ± 0.35	18.99 ± 0.28	16.5 ± 0.32
	**SPD (s)**			
	DIA (LF)	IPSI (RF)	CONT (LH)	L (RH)
G_Sou_	0.44 ± 0.01	0.44 ± 0.01	0.41 ± 0.01	0.41 ± 0.01
G_Cox_	0.48 ± 0.02	0.47 ± 0.02	0.46 ± 0.02	0.44 ± 0.02
	**TPFz%**			
	DIA (LF)	IPSI (RF)	CONT (LH)	L (RH)
G_Sou_	49.33 ± 2.26	48.85 ± 2.67	35.26 ± 1.66	35.53 ± 1.57
G_Cox_	53.1 ± 2.02	52.5 ± 2.14	35.82 ± 1.84	41.84 ± 1.81

G_Sou_ (group sound); G_Cox_ (group coxarthrosis); PFz% (peak of vertical force normalized as percentual of total force); IFz% (vertical impulse normalized as percentual of total force); SPD (s) (stand phase duration in seconds); TPFz% (time of occurrence of peak of vertical force as a percent of the stance phase duration); LF (left front limb); RF (right front limb); LH (left hind limb); RH (right hind limb); DIA (diagonal limb); IPSI (ipsilateral limb); CONT (contralateral limb); L (lame limb).

**Table 3 animals-10-00986-t003:** The *p* values of the PFz%, IFz%, SPD (s), and TPFz% comparing the front and hind limbs within groups.

	Limbs	PFz%	IFz%	SPD (s)	TPFz%
G_Sou_	IPSI vs. DIA (RF vs. LF)	0.56	1.00	0.79	1.00
	L vs. CONT (RH vs. LH)	0.93	1.00	0.98	1.00
	DIA vs. CONT (LF vs. LH)	0.00	0.00	0.00	0.00
	IPSI vs. L (RF vs. RH)	0.00	0.00	0.00	0.00
G_Cox_	IPSI vs. DIA (RF vs. LF)	0.62	0.55	0.13	1.00
	L vs. CONT (RH vs. LH)	0.00	0.00	0.03	0.19
	DIA vs. CONT (LF vs. LH)	0.00	0.00	0.14	0.00
	IPSI vs. L (RF vs. RH)	0.00	0.00	0.00	0.01

vs.: “versus”. Level of significance at *p* < 0.05. G_Sou_ (group sound); G_Cox_ (group coxarthrosis); PFz% (peak of vertical force normalized as percentual of total force); IFz% (vertical impulse normalized as percentual of total force); SPD (s) (stand phase duration in seconds); TPFz% (time of occurrence of peak of vertical force as a percent of the stance phase duration); LF (left front limb); RF (right front limb); LH (left hind limb); RH (right hind limb); DIA (diagonal limb); IPSI (ipsilateral limb); CONT (contralateral limb); L (lame limb).

**Table 4 animals-10-00986-t004:** The *p* values of PFz%, IFz%, SPD (s), and TPFz% comparing limbs between the groups.

Limbs	PFz%	IFz%	SPD (s)	TPFz%
IPSI (RF)	0.04	0.97	0.26	0.30
DIA (LF)	0.15	0.08	0.17	0.22
CONT (LH)	0.54	0.05	0.11	0.82
L (RH)	0.00	0.00	0.35	0.01

Level of significance at *p* < 0.05. PFz% (peak of vertical force normalized as percentual of total force); IFz% (vertical impulse normalized as percentual of total force); SPD (s) (stand phase duration in seconds); TPFz% (time of occurrence of peak of vertical force as a percent of the stance phase duration); LF (left front limb); RF (right front limb); LH (left hind limb); RH (right hind limb); DIA (diagonal limb); IPSI (ipsilateral limb); CONT (contralateral limb); L (lame limb).

**Table 5 animals-10-00986-t005:** The mean values ± standard deviations of PFz%, IFz%, and TPFz% in the CraLa, CraMe, CauLa, and CauMe quadrants of the IPSI (RF), DIA (LF), CONT (LH), L (RH) limbs from the G_Sou_ and G_Cox_.

**PFz%**					
**Groups**	**Quadrants**	**IPSI (RF)**	**DIA (LF)**	**CONT (LH)**	**L (RH)**
G_Sou_	CraLa	9.33 ± 1.07	9.58 ± 1.27	6.13 ± 0.61	5.97 ± 0.64
	CraMe	8.93 ± 1.26	8.76 ± 0.99	5.88 ± 0.70	5.92 ± 0.59
	CauLa	7.53 ± 1.26	7.73 ± 1.20	3.50 ± 0.76	3.57 ± 0.57
	CauMe	5.55 ± 1.03	5.39 ± 0.99	3.20 ± 0.73	3.04 ± 0.70
G_Cox_	CraLa	8.69 ± 1.46	8.63 ± 1.27	5.72 ± 0.9	5.44 ± 1.00
	CraMe	8.69 ± 1.50	8.64 ± 1.41	5.85 ± 1.04	5.54 ± 0.95
	CauLa	8.54 ± 1.80	8.20 ± 1.45	3.85 ± 1.05	3.35 ± 0.70
	CauMe	6.23 ± 1.29	6.12 ± 1.40	3.60 ± 0.92	2.95 ± 0.82
**IFz%**					
	**Quadrants**	**IPSI (RF)**	**DIA (LF)**	**CONT (LH)**	**L (RH)**
G_Sou_	CraLa	10.55 ± 1.16	10.68 ± 0.96	6.99 ± 0.74	6.83 ± 0.84
	CraMe	9.27 ± 1.3	9.06 ± 1.12	6.42 ± 0.98	6.60 ± 0.77
	CauLa	7.07 ± 1.47	7.27 ± 1.35	2.79 ± 0.62	2.78 ± 0.45
	CauMe	4.90 ± 1.25	4.8 ± 1.05	2.05 ± 0.51	1.96 ± 0.50
G_Cox_	CraLa	9.31 ± 1.88	9.54 ± 1.73	6.59 ± 1.16	5.80 ± 1.21
	CraMe	8.60 ± 1.56	8.93 ± 1.77	6.42 ± 1.19	5.76 ± 1.10
	CauLa	8.08 ± 1.42	8.26 ± 1.72	3.31 ± 1.05	2.75 ± 0.61
	CauMe	5.78 ± 1.47	6 ± 1.64	2.67 ± 0.79	2.18 ± 0.67
**TPFz%**					
	**Quadrants**	**IPSI (RF)**	**DIA (LF)**	**CONT (LH)**	**L (RH)**
G_Sou_	CraLa	72.17 ± 6.11	72.84 ± 5.12	63.36 ± 14.24	63.54 ± 11.98
	CraMe	76.95 ± 4.97	77.84 ± 5.01	67.36 ± 13.18	64.46 ± 10.18
	CauLa	37.89 ± 6.84	37.16 ± 5.59	30.38 ± 4.03	30.97 ± 3.83
	CauMe	35.74 ± 9.57	34.77 ± 7.45	28.36 ± 4.06	28.72 ± 3.91
G_Cox_	CraLa	71.54 ± 6.95	72.90 ± 5.69	61.80 ± 14.04	65.57 ± 11.20
	CraMe	76.01 ± 5.74	77.01 ± 5.78	66.02 ± 13.14	66.60 ± 11.06
	CauLa	42.57 ± 9.51	40.92 ± 9.92	31.20 ± 8.34	32.04 ± 7.04
	CauMe	40.10 ± 10.59	39.04 ± 10.92	31.81 ± 9.29	31.87 ± 5.72

PFz% (peak of vertical force normalized as percentual of total force); IFz% (vertical impulse normalized as percentual of total force); SPD (s) (stand phase duration in seconds); TPFz% (time of occurrence of peak of vertical force as a percent of the stance phase duration); LF (left front limb); RF (right front limb); LH (left hind limb); RH (right hind limb); DIA (diagonal limb); IPSI (ipsilateral limb); CONT (contralateral limb); L (lame limb); CraLa (craniolateral); CraMe (craniomedial); CauLa (caudolateral); CauMe (caudomedial).

**Table 6 animals-10-00986-t006:** The *p* values for PFz%, IFz%, and TPFz%, comparing the quadrants CraLa, CraMe, CauLa, and CauMe of the IPSI (RF), DIA (LF), CONT (LH), L (RH) limbs within G_Sou_ and G_Cox_.

**Within G_Sou_**		***p* Values**		
**Limbs**	**Quadrants**	**PFz%**	**IFz%**	**TPFz%**
DIA (LF) vs. IPSI (RF)	CraLa	0.49	0.68	0.69
	CraMe	0.61	0.56	0.55
	CauLa	0.58	0.64	0.68
	CauMe	0.60	0.77	0.70
CONT (LH) vs. L (RH)	CraLa	0.38	0.50	0.96
	CraMe	0.83	0.49	0.41
	CauLa	0.72	0.97	0.62
	CauMe	0.45	0.57	0.76
**Within G_Cox_**		***p* Values**		
**Limbs**	**Quadrants**	**PFz**	**IFz**	**TPFz**
DIA (LF) vs. IPSI (RF)	CraLa	0.87	0.68	0.47
	CraMe	0.92	0.50	0.56
	CauLa	0.47	0.69	0.57
	CauMe	0.78	0.64	0.74
CONT (LH) vs. L (RH)	CraLa	0.32	0.03	0.32
	CraMe	0.30	0.06	0.87
	CauLa	0.06	0.04	0.71
	CauMe	0.02	0.03	0.98

vs.: versus. Level of significance at *p* < 0.05. PFz% (peak of vertical force normalized as percentual of total force); IFz% (vertical impulse normalized as percentual of total force); SPD (s) (stand phase duration in seconds); TPFz% (time of occurrence of peak of vertical force as a percent of the stance phase duration); LF (left front limb); RF (right front limb); LH (left hind limb); RH (right hind limb); DIA (diagonal limb); IPSI (ipsilateral limb); CONT (contralateral limb); L (lame limb); CraLa (craniolateral); CraMe (craniomedial); CauLa (caudolateral); CauMe (caudomedial).

**Table 7 animals-10-00986-t007:** The *p* values for PFz%, IFz%, and TPFz% comparing the quadrants CraLa, CraMe, CauLa, and CauMe of the IPSI (RF), DIA (LF), CONT (LH), L (RH) limbs between G_Sou_ and G_Cox_.

Limbs	Quadrants	PFz%	IFz%	TPFz%
Ipsilateral (RF)	CraLa	0.10	0.01	0.75
	CraMe	0.55	0.12	0.56
	CauLa	0.03	0.02	0.06
	CauMe	0.06	0.03	0.15
Diagonal (LF)	CraLa	0.01	0.01	0.97
	CraMe	0.74	0.77	0.61
	CauLa	0.25	0.03	0.12
	CauMe	0.05	0.01	0.13
Contralateral (LH)	CraLa	0.08	0.17	0.71
	CraMe	0.91	0.98	0.73
	CauLa	0.21	0.04	0.68
	CauMe	0.11	0.00	0.11
Lame (RH)	CraLa	0.04	0.00	0.56
	CraMe	0.11	0.00	0.50
	CauLa	0.23	0.88	0.52
	CauMe	0.69	0.22	0.03

PFz% (peak of vertical force normalized as percentual of total force); IFz% (vertical impulse normalized as percentual of total force); TPFz% (time of occurrence of peak of vertical force as a percent of the stance phase duration); LF (left front limb); RF (right front limb); LH (left hind limb); RH (right hind limb); CraLa (craniolateral); CraMe (craniomedial); CauLa (caudolateral); CauMe (caudomedial).

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
