# Peer review of "Comparison of the Vertical Force Distribution in the Paws of Dogs with Coxarthrosis and Sound Dogs Walking over a Pressure Plate"

_animals, 2020, doi:10.3390/ani10060986_

Round 1
Reviewer 1 Report
The authors report the comparison of the Vertical Force Distribution on Pads of hip osteoarthrosic dogs with sound dogs. This a very interesting and well-designed study.
I have only minor revisions to suggest:
Title
I suggested this variation for the title: Comparison of the Vertical Force Distribution in the Pads of Dogs with Coxarthrosis and Sound Dogs by the use of a Pressure Plate
Simple summary
Lines 16-17: please use a correct anatomic reference for “caudal region”. Caudal quadrants?
Abstract:
Line 21: please delete (GSou) and (GCox). Remember to revise the sentences of the abstract that include these abbreviations.
Lines 21-22: “ The dogs walked over…”
Line 30: change “hind leg” with “hindlimb”
Introduction
Lines 35-36: change “ Thus, in the case of hip arthrosis, the development..” with “ Thus, when a coxarthrosis occurs, the development..
Line 41: change “There are also kinetic and kinematic studies, which help to..” with “ Kinetic and kinematic studies help to…”
Line 45: Change “sick animals” with “lame animals”.
Line 45: Change “The most commonly used techniques are force..” with “The most common techniques for gait analysis are: force..”
Line 50: Delete “In general, the”
Line 52: “redistribute the load of body weight”.
Line 56: Change “They found, via Fourier analysis, that a unilateral” with “Those authors found that an unilateral..”
Lines 58-59: please reformulate this sentence. It is not clear.
Line 63: Change “…force between the paws would change compared to sound animals and the non-affected limbs.” With “…force between the paws would change comparing the non-affected limbs of diseased dogs with same paws of sound dogs.” Have I correctly interpreted the meaning of your sentence?
Materials and Methods
Lines 75-77: Revise as follow: “In the GSou there were six females, nine males, four spayed females, and three neutered males, with a mean age of 4.3 ± 2.3 years and a mean body mass (BM) of 25.4 ± 9.1 kg. To determine their good health status, the dogs were undergone to physical, neurological, and orthopedic examinations. “
Lines 80-81: “according to….animals” This sentence is not clear, please reformulate it
Line 84:” Vizsla)”
Lines 84-86: Change “Of which, there 84 were one female, six males, eight spayed females, and seven neutered males), aged 8.5 ± 3.3 years, 85 with a mean BM of 27.5 ± 11.6 kg,..” with “In the Gcox, there were one female, six males, eight spayed females, and seven neutered males, with a mean age of 8.5 ± 3.3 years, with a mean BM of 27.5 ± 11.6 kg,..”
Line 91: “present a SI%”.
Lines 97-98: Please reformulate this sentence “The result is given in percent and the closer to zero, the higher the symmetry between limbs.”. It is not clear
Results
Lines 144-146: This is a repetition of next part! Pay attention!
Line 154: “3.2.1. Within Groups” Please present better next part with appropriate title paragraph.
Line 162: “L” I do not understand this abbreviation. Left? Lame?
Line 225: Table 6: Please report in legends how significance was underlined.
Discussion:
Line 299: “From this,..” this? present study? I do not understand
Line 323: Change “..similar to sick animals and…” with “..similar to dogs of Gcox and…”
Line 336 “we demonstrated that..” We? is this data referred to your investigation or to Souza's study? This sentence is not clear.
Figures and Tables:
Please pay attention that all abbreviations have been correctly explained within the legends.
I should suggest a change of colours (not only grey scale) to better improve the contrast and visibility of the figures.
References
I should suggest also to insert this reference as study that compared force plate and pressure pad techniques for gait analysis:
Sandberg G, Torres B, Berjeski A, Budsberg S. Comparison of Simultaneously Collected Kinetic Data with Force Plates and a Pressure Walkway. Vet Comp Orthop Traumatol. 2018 Sep;31(5):327-331. doi: 10.1055/s-0038-1666875.
Reviewer 2 Report
Thank you for submitting this interesting manuscript dealing with the study of the load distribution during the walk of dogs with coxarthrosis. The authors show relevant data concerning to load distribution within limbs and an interesting comparison between sound dogs and dogs with coxarthrosis have been made. The data the authors show are numerous and of great interest and I want to encourage the authors to put effort into revising their manuscript to make it stronger. As the authors stated, the most common cause of chronic pain in dogs is osteoarthrosis. Then, such relevant data are very much needed by veterinary clinicians. After reviewing the manuscript, I would like to present the following comments. Some of them need thorough revision, and others are suggestions and comments that I believe may improve the manuscript.
General comment:
Reconsider the terminology used for the variables (parameters): (SI (%PFz)) PFz (%TF), IFz (%TF), and, TPFz (%SPD). Decide terms to use and change throughout the manuscript.
Reconsider the terminology limb/leg/member. Decide terms to use and change throughout the manuscript.
Title:
Consider changing from “pads” to “paws”. You studied the load distribution in the distal part of the limb, the fore and hind paws, not in the digital and metacarpal/metatarsal pads. I also suggest adding “walking” before “dogs. This study was carried out in walking dogs.
Simple summary:
Line 15: Remove “pads”
Line 16: Change from “animals” to “walking dogs”. The study was in walking dogs and results may have been quite different in trotting dogs.
Line 21: I think they are 23 dogs in the two groups. Revise the number of dogs.
Keywords: Please rearrange them in alphabetic order.
Introduction:
Line 44: consider changing from “sick animals ” to “ “dogs with orthopaedic conditions”.
Line 45: consider changing from “healthy” to “sound”
Line 54-55: change “in dogs” to “in trotting dogs” and from “within the” to “to”.
Line 58: change to “member” to “limb”. And use it throughout the manuscript.
Line 62: change from “quadrant” to “cranial/caudal and lateral/medial”. At the moment the term “quadrant” has not been defined.
Material and Methods:
General comment: This part needs a major revision. Parameters are not properly defined and the terminology needs to be reconsidered. However, I want to encourage the authors to thoroughly review this and other parts of the manuscript. I am keen on getting this manuscript through the review process and would be happy to see it published.
Line 72: change from “GSou (group sound)” to “group sound (GSou)”
Line 73 and 82: In both groups are 23 dogs
Line 82: change from “GCox (group coxarthrosis)” to “group coxarthrosis (GCox )”
Line 88: One of the criteria of inclusion is hindlimb lameness. In one or in both hidlimbs? Mostly in right hindlimb?. This must be clarify.
Lines 91-98: I recommend adding the “symmetry index” like a parameter to be described afterwards together with other parameters. However, if you have decided to include the symmetry index of peak vertical force and of vertical impulse among the criteria of inclusion you must state this in this paragraph. Besides, It is unclear what is the level of symmetry index to be included, this need to be clearly expressed in this paragraph. All calculations of the symetry index should be with the description of the variable, in the “Measurement Procedures” section.
Line 104: consider changing from “constant velocity” to “regular and/or comfortable velocity”. You did not measure the velocity throughout the whole movement, you get the mean velocity and aceleration.
Line 106: change from “passages” to “trials”. What was the function of the recording? It should be indicated.
Line 109-115: As I mention previously my concern is that the parameters (variables) to be measured are not clearly described, defined and named. Abbreviations are too long and lead to confusion. Please one by one must be named, abbreviated, and defined. Besides, as recommended above, the symmetry index should be included here like other parameters.
I suggest shorter terms:
- SI%: define and explain how it is calculated.
- PFZ%: peak of vertical force as the percentage of the total force. Measured in Newton.
- IFz%: vertical impulse as the percentage of the total force. Measured in Newton/seg.
- TPFz%: time of occurrence of peak of vertical force as a percent of the stance phase duration.
Line 113: Why the right hind limb is referred as “lame”. As you stated above, dogs with coxarthrosis in “at least one hip articulation”, lameness (you did not mention if the dogs were lame more from the right hind limb). You also mentioned that the symmetry index (SI%) was also determined as an inclusion criterion. However, you did not mention the values of the symmetry index to classify dogs lame from the right hindlimb or mainly lame from the right hindlimb. Authors are encourage to stablish clearly the criterion and values to classify dogs within Gcox.
Line 118: Remove all the sentence. The parameters have already described.
Line 119: Remove all the sentence. The parameters have already described.
Line 126: “Values were normalize”. Normalize according to what??
Results
Line 148-150: Consider change the order of the sentences. First present the table and afterwards the results, and afterwards include the table.
Line 152: change from “different letters” to “different superscript”. The table legend does not offer enough information to the reader. Please, add data to improve it in all the tables and figures.
Line 153: Consider adding a brief sentence to explain what the following section is going to show.
Line 166-168: This sentence should be before the results, at the beginning of the line 155.
Line 162: consider adding “in CONT” after “higher”
Line 163: consider adding “in CONT” after earlier.
Line 165-166: change from “were statistically equal” to “were not statistically different”
Line 166: SPD (s). Remove (s). It is the unit. Express above the unit and omit throughout the text. It is not necessary.
Figure 3 and 4: Differences between groups are not clear. Please, add a clear explanation.
Figures 3 and 4 should be after the paragraph where the differences between groups have been shown.
Line 182: change from “In lame limbs (i.e., RH of GSou)” to “In right hindlimbs of sound dogs (RH, Gsou)”
Line 183: remove Gsou.
Line 186-187: The sentence should be at the beginning of the paragraph.
187: add “numerically” after “presented”.
Line 192: Present here the table 5 before the results, better than in line 219.
Line 199: “both compared to the healthy group”. Be careful, the previous sentence is referred to the Gcox.
Line 208: “ a compensatory…”. This statement sould be in the Discussion section.
Line 213: “compensating..”. It is also for the discussion section.
Line 218: change from “lame limb” to “L”.
Line 228 and 233: there is not “reduction/increase” because the dogs are not the same. “values are higher or lower in… than in ….”
Line 230: “compensating……..lame limb”. Better in the discussion section.
Line 243: “These results are summarized in Figures 5–7, and the p values are listed in Table 7.”. The sentence should be before showing the results, at the beginning of the paragraph. In line 236.
Discussion
Please revise the terminology of the parameters measured and use all along the manuscript.
Line 305: Consider changing from “by Pitbulls” to “in Pitbulls”
Line 310: “member”. Please be consistent with the terminology that you indicated at the beginning of the manuscript.
Line 310 -: The sentence “There was a clear redistribution of the load to the unaffected member (CONT), both when comparing groups and comparing quadrants within GCox itself” is not clear. Consider to rewrite in order to a better undersatanding.
Line 323: “morphologically similar”. This is not true. You have dogs as different as Dachshund and Rottweilers. Besides, the standard deviation of the body mass of the Gsou is quite high. Differences in breed and in body mass indicate that they are different morphologically. This must be conveniently cited as a limitation of the study.
Line 324: Please consider changing from “healthy” to “sound”.
Line 328 : the sentence “It is difficult …..limb” is not related with the study. You have studied dogs with coxarthrosis and sound dogs, and the differences between them may be attributed to the hip coxarthrosis.
Line 336: “we demonstrated”. Please, rewrite the sentence referring to the authors of the study.
Line 350: please consider changing from “healthy” to “sound”.
